# GraphCSR: A Space and Time-Efficient Sparse Matrix Representation for Web-scale Graph Processing

## ABSTRACT

Graph data processing is essential for web-scale applications, including social networks, recommendation systems, and web of things (WoT) systems, where large, sparsely connected graphs dominate. Traditional sparse matrix storage formats like compressed sparse row (CSR) face significant memory and performance bottlenecks in distributed, federated, and edge-based computing environments, which are increasingly central to the web. To address this challenge, we propose GraphCSR, a novel storage format that clusters vertices with identical edge degrees and stores only the starting index of each group. This approach minimizes memory overhead and facilitates batch memory access while enhancing overall performance, making it particularly suitable for federated systems and resource-constrained edge nodes. Our experiments across various graph operations and large datasets show that GraphCSR achieves considerable memory savings and performance gains of large-scale, distributed graph processing. When deployed GraphCSR on a production-grade supercomputer with 79,024 computing nodes, it outperforms the top-ranked system on the Graph 500 list, demonstrating its potential for scaling web and WoT graph processing in large-scale distributed computing systems.

## KEYWORDS

Web-scale graph processing, Sparse matrix storage, CSR, Distributed graph analytics

## 1 INTRODUCTION

Graph processing is a cornerstone of many web-scale applications, playing a vital role in modern services like social networks [10, 27], recommendation systems [41, 54], and the WoT [26, 49]. These applications rely on graph data to model relationships between entities, enabling powerful features such as user connectivity, content recommendations, and device communication. For instance, social networks use graph structures to represent users and their interactions, allowing platforms to identify key influencers or detect communities within vast networks. Similarly, recommendation systems leverage graph processing to analyze user-item interactions, driving personalized content delivery that enhances user experience. In the context of WoT, graph data models interactions between interconnected devices, enabling efficient device communication, resource management, and automation in smart environments. Operations like breadth-first search (BFS) and depth-first search (DFS) allow systems to traverse these complex data structures, uncovering valuable patterns such as connectivity, shortest paths, or clusters. These capabilities are foundational in enabling web applications to derive insights from vast amounts of data, ensuring services remain responsive and adaptive as user bases and data volumes grow.

In graph processing, the relationship between vertices in a graph is usually represented using an adjacency matrix. In this matrix, an element $a_{u,v}$ is set to one if an edge exists between vertices $u$ and $v$; otherwise, the element is set to zero. Due to the sparsity of many real-word graphs, where most vertices are low-degree vertices with only a small number of edges connected to them, the adjacency matrix is inherently sparse [6, 8, 12, 52]. As such, the adjacency matrix is typically stored in a sparse matrix format to reduce the memory footprint and improve the computational efficiency of graph processing algorithms.

The CSR format is extensively utilized for representing sparse matrices in the context of web-scale graph processing [8, 37, 47, 52]. CSR focuses on recording only the non-zero entries of a matrix, supported by auxiliary information such as row pointers and column indices. Unlike traditional sparse matrix formats, which are primarily designed to optimize sparse matrix-vector (SpMV) or sparse matrix-matrix (SpMM) multiplications [13, 33, 37] and improve performance on heterogeneous hardware [25, 38], CSR is specifically crafted to enhance storage efficiency. This characteristic makes it especially advantageous for large-scale graph traversal algorithms [36, 39, 47, 52], where memory constraints can present significant challenges.

However, the increasing scale and complexity of web applications introduce significant challenges in processing these graphs efficiently, particularly in distributed, federated, and edge-based environments where resources like memory and bandwidth are limited [5, 17, 30, 32, 34, 48, 53]. Traditional sparse matrix storage formats, such as the CSR and its variants [8, 12, 47], often fall short in these contexts, as they do not fully exploit the characteristics of real-world web-scale graphs. Many of these graphs are sparse and exhibit skewed degree distributions [23, 31], where most vertices have few edges. The uniform handling of vertices, by storing both low- and high-degree vertices as individual non-zero entries, results in inefficiencies, as low-degree vertices unnecessarily contribute to memory overhead and exacerbate bottlenecks in large-scale graph processing. Addressing these challenges requires innovative storage and computation strategies to handle the unique demands of web-scale graph processing, laying the groundwork for more optimized and scalable solutions.

We present GraphCSR[1], a groundbreaking sparse matrix storage format specifically designed to enhance memory efficiency for web-scale graphs with skewed vertex degree distributions. Building upon the widely adopted CSR format, GraphCSR requires minimal modifications to existing graph algorithm implementations, making it easy to adopt.

Our central insight is that many graph vertices tend to exhibit the same low-edge degree, allowing for effective grouping that significantly reduces storage needs and runtime memory consumption. GraphCSR capitalizes on the graph sorting step commonly utilized in most parallel graph processing algorithms [11, 18, 22, 23, 35, 36, 56] by first organizing vertices based on their edge degrees. Within each grouped category, GraphCSR compresses data by only

---

[1]Code available at *https://anonymous.4open.science/r/GraphCSR-450E/README.md*

recording the starting vertex, thus minimizing the overall memory footprint for graph processing tasks. During execution, GRAPHCSR employs a straightforward formula to efficiently check for edge connections between any pair of vertices, facilitating standard graph processing operations while simultaneously reducing memory usage. By grouping vertices with the same degree, GRAPHCSR also enables batching and coalescing of memory accesses, thereby enhancing the performance of web-scale graph processing.

We evaluate GRAPHCSR in the context of web-scale graph processing by applying it to key graph algorithms on widely-used, large-scale graph datasets [19, 51, 57, 58]. As baselines, we compare against eight prominent sparse matrix storage formats [8, 12, 18, 22, 23, 47, 52, 56]. Our evaluation is conducted on a large-scale HPC system, utilizing up to 79,024 nodes and over 1.2 million processor cores to simulate web-scale environments. Experimental results consistently demonstrate the superiority of GraphCSR, providing greater storage efficiency and faster processing times compared to all baselines. When tested on the Graph 500 BFS benchmark, GRAPHCSR outperforms the highest-ranked supercomputer on the latest Graph 500 list (June 2023), achieving a 1.6× increase in throughput, while reducing memory consumption by 25% and using fewer CPU cores.

The main contribution of this paper is a novel sparse storage format optimized for web-scale graph processing. Our theoretical analysis highlights the format's efficiency in handling massive graph datasets common in web-scale systems, while the empirical results confirm its performance advantages over state-of-the-art sparse matrix storage methods in distributed graph environments.

This paper presents several significant contributions to the field of web-scale graph processing:

- An important insight is that reordered *CSR* format can be effectively leveraged in Web-scale graph processing to store and index vertices with identical degrees, particularly low-degree vertices, in sorted graphs. This approach enhances space-time efficiency, making it ideal for managing large-scale graph data in distributed environments.

- We showcase how to exploit the characteristics of low-degree vertices in sorted graphs to enhance CSR-like formats for web-scale graph processing. This improvement facilitates the execution of essential graph algorithms and SpMV operations, enabling efficient graph analysis on large-scale datasets in distributed systems.

- Extensive experiments demonstrate that GRAPHCSR achieves rapid processing times and a reduced memory footprint compared to state-of-the-art sparse matrix storage formats in web-scale graph processing for distributed graph applications. Notably, GRAPHCSR ranks at the top of the Graph500 BFS leaderboard, handling up to 77.2K nodes with a 57.8% increase in throughput and a 74% reduction in memory usage. These results highlight the effectiveness and efficiency of GRAPHCSR in managing large-scale graph datasets.

## 2 BACKGROUND AND MOTIVATION

### 2.1 Graph Processing

As depicted in Figure 1, web-scale graph processing generally consists of three key stages: (i) graph preprocessing, (ii) graph construction, and (iii) graph computation. During the preprocessing stage, various tasks like vertex isolation and pruning, degree counting, and vertex sorting are performed. Note that many large-scale graph processing algorithms and parallelization strategies require sorting the graph vertices according to the edge degree at the preprocessing stage [18, 20, 22, 23, 36, 45, 56]. In the graph construction stage, vertices and edges are stored in specific storage formats [14, 20, 28, 45, 52], such as CSR, or bitmap. Finally, the constructed graphs are passed into a graph processing algorithm like BFS, Single Source Shortest Path (SSSP), Connected Component (CC), PageRank (PR), Betweenness Centrality (BC), Triangle Counting (TC) [24, 56] and SpMV. This structured approach ensures efficient processing of large-scale graph data across distributed systems.

**Problem scope.** Our work focuses specifically on the graph construction stage within the context of web-scale graph processing, aiming to optimize the representation of the graph adjacency matrix. This optimization is crucial for reducing the memory footprint, a significant challenge in large-scale, distributed graph processing environments. Once the graph is constructed, GRAPHCSR is equipped to support various graph computations, including SpMV operations (see section 5.6), enhancing the overall efficiency of processing massive graphs in web, federated, and edge-based environments. This capability ensures scalability and performance across distributed systems central to modern web-scale applications.

**Graph partitioning.** In the realm of web-scale graph processing, two predominant strategies for partitioning graphs and distributing the elements of the graph adjacency matrix for parallel processing are 1-dimensional (1D) and 2-dimensional (2D) decomposition [9]. 1D graph decomposition involves breaking down a graph adjacency matrix into smaller sub-matrices, each of which can be owned by a processor independently. A graph is often represented as a sequence of vertices and edges, and the decomposition process involves partitioning this sequence into smaller sub-sequences or components. This approach is widely used in graph processing and has been extensively studied [42–44]. In contrast, 2D graph decomposition represents a graph as a mesh or an edge-vertex matrix, which can be partitioned into smaller sub-matrices and assigned to each processor for distributed processing. This approach has also been extensively studied and has shown promising results in recent researches [22, 23, 56]. In this work, our discussion is centralized on 2D graph partitioning, but our techniques can be equally applied to 1D decomposition.

### 2.2 CSR for Graph Representation

The CSR format is a widely adopted representation for storing sparse matrices in the context of web-scale graph processing. Compared to other sparse matrix formats like DOK (Dictionary of keys), LIL (List of lists), or COO (Coordinate list) [16, 33, 37], CSR offers efficient access and matrix operations and yields a better storage solution. Specifically, CSR represents a sparse matrix using three one-dimensional arrays: val, RST, and COL. The val array stores the non-zero values in contiguous locations, with each row packed together. The RST array specifies the starting index of each row in the val array, while the COL array maps each non-zero value to its corresponding column. Graph processing algorithms mostly focus on accessing each vertex and then filtering vertices on demand. For clarity, the val array can be omitted for better understanding.

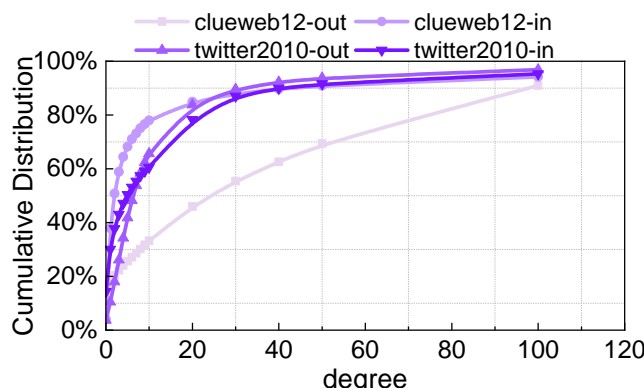

Figure 1: A typical graph processing pipeline.

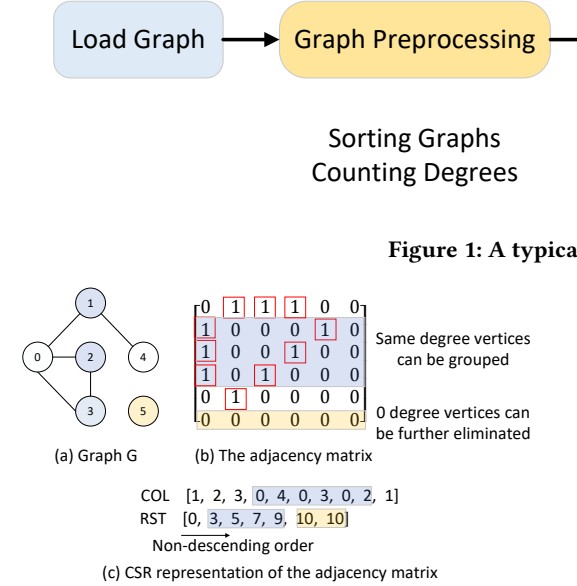

Figure 2: CSR format (c) of the graph adjacency matrix (b) for a graph G (a).

However, in web-scale graph processing, where datasets are immense and constantly evolving, the limitations of traditional CSR become more pronounced. The need for rapid access, scalability, and efficient memory usage is critical. Graph processing frameworks that are designed for web-scale applications often utilize alternative storage formats or methods (like adjacency lists or distributed graph databases) that can handle large-scale datasets more effectively and allow for more efficient parallel processing.

Many real-life graphs exhibit sparsity and unevenness[2]. General-purpose sparse matrix storage formats [8, 12, 47, 52] only focus on removing 0-degree vertices for better memory usage. However, GraphCSR is tailored for graph processing by further compressing the graph adjacency matrix by grouping the same-degree vertices (i.e. $vertex\{1, 2, 3\}$ shown in Figure 2). It is worth noting that same-degree vertices with low degrees constitute the vast majority of vertices in both real-world and synthetic graphs[22]. As a result, there is lots of room for GraphCSR to demonstrate its potential.

Many real-world graphs are characterized by sparsity and unevenness, often described by the small-world model, where most vertices have a low degree and are connected to only a few high-degree vertices. This structure is significant in the context of web-scale graph processing, as efficient management of large-scale data becomes crucial. Traditional sparse matrix storage formats primarily focus on removing 0-degree vertices to optimize memory usage. However, our approach, GraphCSR, is specifically designed for graph processing at scale by further compressing the graph adjacency matrix. It achieves this by grouping same-degree vertices (e.g. $vertex\{1, 2, 3\}$ shown in Figure 2). Notably, the majority of vertices in both real-world and synthetic graphs possess low degrees, indicating that these same-degree vertices represent a substantial

---

[2]This is also known as a small world model - where most vertices of a real-life graph are low-degree vertices with only a small number of edges connected to very few high-degree vertices [15, 22].

Figure 3: Degree cumulative distribution of clueweb12 and twitter-2010, such that graph-in/out represents the in-degree or out-degree of the graph.

portion of the graph. This opens up significant opportunities for GraphCSR to enhance performance and efficiency in processing large graphs, making it particularly valuable for applications that require scalable and efficient graph analysis.

## 2.3 Motivation

Our work is built upon two key observations regarding the row index array, RST when using CSR to store the adjacency matrix of a sorted graph. Firstly, the array elements are ordered in a non-decreasing manner, as exemplified by the sequence 0, 3, 5, 7, 9, 10, 10 in Figure 2(c), and some values in RST may only refer to rows that correspond to 0-degree vertices, where values are all zero. For example, in Figure 2(c), the last two elements, 10, 10, of RST, refer to the last row of the adjacency matrix in Figure 2(a), which shows the connected edge of the 0-degree vertex 5 in Figure 2(a).

Secondly, vertices with the same edge degree have their adjacent lists stored in a contiguous manner in the COL array. For the example given in Figure 2, the 2-degree vertices $v_1$, $v_2$ and $v_3$ in the COL array are represented by the sets $\{0, 4\}$, $\{0, 3\}$ and $\{0, 2\}$ respectively.

Besides clueweb12 and twitter-2010, we conduct tens of public graphs and get similar distributions, in which low-degree vertices make up a large proportion of real-life graphs, thereby occupying a substantial portion of the storage space. Figure 3 shows how the vertices distribute in two real-world graphs - clueweb12 and twitter-2010. While graph clueweb12 owns 978,408,098 vertices and has a sparsity (i.e., non-zero degree vertices ratio) of 90.5%, graph twitter-2010, on the other hand, owns 41,652,230 vertices with the sparsity of 86.7%. In Figure 3, we show that 0-degree vertices account for up to 9.5%, vertices with $0 \le degree \le 9$ constitute approximately

78%, the proportion of vertices ( $0 \leq degree \leq 20$) is steadily increases to 85%, while the ratio of vertices with $0 \leq degree \leq 50$ is close to 91% but incrementally climbs to 95% for vertices with $0 \leq degree \leq 100$. Interestingly, the distribution of low-degree vertices does not vary significantly with the sparsity of the graph though the number of zeros decreases as the graph gets denser. For example, even a denser graph like clueweb12 (whose average degree is 43.5) has less than 10% 0-degree vertices, it still holds 78% of the vertices with $0 \leq degree \leq 9$. In addition, [21] showed more details on the distribution of low-degree vertices. This scenario adds an extra dimension of motivation to develop a new representation for large-scale graphs that contain a large proportion of low-degree vertices. In the context of web-scale graph processing, various CSR-like formats have been introduced to tackle the challenge of storing rows that consist entirely of zero values. Notable examples include DCSR [8], CSCSR (Coarse Index + Skip List) [12], and BCSR (Bitmap-based Sparse Matrix Representation) [47]. While these techniques effectively address the issues raised by our initial observation, they fail to capitalize on the potential benefits of grouping vertices with the same degree. This grouping could substantially minimize the memory requirements when processing large-scale graphs. To address this shortcoming, GRAPHCSR is specifically designed to optimize memory usage and enhance efficiency in web-scale graph processing scenarios.

The primary goal of GRAPHCSR is to enhance memory efficiency and optimize access patterns for large-scale graph algorithms, specifically tailored for web-scale graph processing. By effectively grouping low-degree vertices that share the same degree, GRAPHCSR facilitates batched memory accesses, allowing a single load operation to retrieve multiple adjacent lists of vertices with identical edge degrees simultaneously. This optimization is achieved by storing only the starting addresses of the adjacency lists for vertices with the same degree, resulting in a significant reduction in memory usage. Such space savings are crucial when processing large graphs, where graphs are typically dominated by low-degree vertices. Traditional sparse matrix storage formats often overlook the structural characteristics of such large, sorted graphs, leading to inefficiencies. In contrast, GraphCSR directly addresses this gap, improving memory utilization and enhancing performance for processing massive graph datasets in distributed, federated, and edge-based environments, key to modern web and WoT systems.

## 3 OUR APPROACH

In this section, we use BFS as a working example to explain GRAPHCSR, but GRAPHCSR can be used with any other graph processing algorithm to represent sparse matrices.

### 3.1 Overview of GRAPHCSR

GRAPHCSR enhances the traditional CSR format by utilizing a folding method to consolidate vertices with the same degree into a single starting offset in the RST array. This straightforward yet powerful folding scheme is particularly effective for web-scale graph processing, as many real-world graphs contain a significant number of low-degree vertices that share identical edge degrees. By optimizing the storage and access patterns in this manner, GRAPHCSR

significantly improves the efficiency of graph algorithms, making it well-suited for handling the massive and complex datasets typical in web-scale applications.

GRAPHCSR is particularly effective when applied to sorted graphs generated during the preprocessing stage of many parallel graph processing algorithms, which are essential for handling large-scale datasets. Since numerous distributed graph processing algorithms require the sorting of graph vertices based on their edge degrees, the sorted graph's adjacency matrix is often readily available during construction . In these sorted graphs, vertices with the same degree are stored contiguously in the RST array of CSR format. GRAPHCSR leverages this structure to reduce memory usage by storing only the initial vertex ID within each group of vertices sharing the same degree. This optimization not only improves the coherence of vertex access patterns but also significantly reduces the time needed for vertex traversal. During graph computation, a straightforward calculation allows for the determination of vertex IDs of the folded vertices using the starting index of the relevant vertex group and the starting index of the nearest edge degree group, enhancing the efficiency of processing large-scale graphs commonly encountered in web applications.

### 3.2 The GRAPHCSR Sparse Storage Format

Specifically, GRAPHCSR introduces two additional arrays, namely `low_deg_RST` and `low_deg_COL`, and a hyperparameter, denoted as `Thr`. The `Thr` is a threshold that determines whether a vertex is considered low-degree and should be folded. The `low_deg_RST` array stores the number of low-degree vertices, while `low_deg_COL` array stores the COL offset value of a vertex concerning the lowest ID of a group of vertices with the same edge degree. It is worth noting that the `low_deg_COL` array points to the original COL array and serves as an offset of the COL array. This way, the low-degree vertices can be traced through the `low_deg_COL` array. For high-degree vertices whose edge degrees are greater than `Thr`, we store them in the standard CSR RST and COL arrays. In contrast to low-degree vertices, GRAPHCSR chooses not to compress high-degree vertices. This decision is based on the observation that high-degree vertices are usually relatively rare in real-world graphs and are often accessed frequently during graph traversals. Therefore, compressing them may incur additional runtime overhead, potentially outweighing the benefits of compression. In GRAPHCSR, we refer to the RST and COL arrays used to store high-degree vertices as `high_deg_RST` and `high_deg_COL`, respectively, to aid clarity.

### 3.3 Graph Construction

We use Figure 4 as a working example to illustrate how GRAPHCSR represents a graph adjacency matrix for the example given earlier in Figure 2. For illustration, we set the edge-degree threshold parameter, `Thr`, to 2. This means that vertices with an edge degree greater than 2 are considered high-degree vertices and stored in the `high_deg_RST` array, while vertices with a degree equal to or less than 2 are considered low-degree vertices and stored in the `low_deg_RST` and `low_deg_COL` arrays by referring to the COL array.

Algorithm 1 outlines how GRAPHCSR encodes high-degree and low-degree vertices. Specifically, the `low_deg_RST[N]` stores the

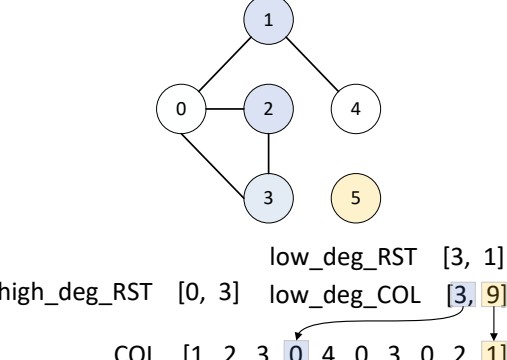

Figure 4: Using GraphCSR (Thr =2) to represent the graph adjacency matrix of Figure 2(b). Here, low_deg_RST[N] records the number of $(Thr − N)$-degree vertices, such that $N ∈ [0, Thr)$, and low_deg_COL[M] records a starting offset in COL for the vertex with $(Thr − M)$-degree, such that $M ∈ [0, Thr)$.

---

**Algorithm 1:** GraphCSR constructing algorithm

**Input:** RST
         Thr // Degree Threshold
**Output:** *low_deg_RST*
         *high_deg_RST*
         *low_deg_COL*

  // Initialization
1  *low_deg_RST ← NULL*
2  *high_deg_RST ← NULL*
3  *low_deg_COL ← NULL*
4  **for** $v_c ∈ [0, len(RST)\text{-}1]$ *in parallel* **do**
5      degree = RST[$v_c$+1]-RST[$v_c$]
6      **if** *degree ≤ Thr* **then**
7         *low_deg_RST[Thr − degree]++*
8         *low_deg_COL[Thr − degree] = RST[$v_c$]*
9      **else**
10        *high_deg_RST.append(RST[$v_c$])*

---

number of $(Thr − N)$-degree vertices for $N ∈ [0, Thr)$, with the edge degree ordered in a descending manner[3]. For example, in Figure 4, low_deg_RST[0] equals 3 because there are three vertices, $v_1$, $v_2$ and $v_3$, with an edge degree of 2, and low_deg_RST[1] equals 1 because there is one vertex, $v_4$, with an edge degree of 1. The low_deg_COL[M] represents a starting offset in COL for the vertex with $(Thr − M)$-degree, with $M ∈ [0, Thr)$. Finally, the high_deg_RST array is the standard CSR RST array, but it only stores vertices with degrees greater than Thr. Overall, this approach optimizes the storage of low-degree vertices while still allowing high-degree vertices to be stored using the standard CSR approach.

## 4 EXPERIMENTAL SETUP

### 4.1 Evaluation Platforms & Workloads

To evaluate the portability of GraphCSR, we apply it to two HPC systems with different CPU architectures using 512 nodes and 8,192 processor cores. Table 2 lists the two HPC systems used in our testing and the maximum number of computing nodes used in experiments. Each node on the WuzhengLight has two HG2 32-core CPUs at 2.5 GHz that are compatible with the AMD x64 instruction set. Each node on Tianhe-Exa has a Phytium 16-core CPU at 2.0 GHz. Both systems run a customized Linux operating system with Linux kernel v9.3.0. We use MPICH 10.2.0 for the message passing interface (MPI) and libgomp 4.5 for OpenMP. We compile the benchmark using GCC 10.2.0 with "-O3" as the compiler option.

Our main evaluation is performed on the BFS algorithm defined in the Graph500 benchmark [28]. Graph 500 is the *de facto* standard for assessing a computer system's capability for graph processing [22, 35, 36, 40, 46]. It provides a graph generator to generate synthetic graphs that mimic real-life graph structures. This tool takes two parameters, a graph factor, and an *edge_factor*. For a graph size $m$ and an *edge_factor* $n$, the generator generates a graph of $2^m$ vertices and $n × 2^m$ edges. Unless stated otherwise, we use the Graph 500 default edge factor of 16.

In addition to synthetic graph data generated by Graph500, we also evaluate our GraphCSR on two public graphs collected from real-life social networks [2, 3], including clueweb12 (with 987 million vertices and 42.6 billion edges) [3], and twitter-2010 (with 41.7 million vertices and 1.47 billion edges) [2] along with other graph processing operations, including DFS, SSSP, PR, CC, BC, and TC.

### 4.2 Evaluation Methodologies

In this paper, we adhere to the Graph 500 ranking methodology to evaluate the throughput of graph processing. To quantify this, we report the giga-traversed edges per second (GTEPS), which is a *higher-is-better* metric. For each test case, we conduct ten runs on unloaded machines, randomly selecting 64 root vertices for each run. We then report the geometric mean of the results. It's worth noting that the variances of GTEPS across different runs for the same test case are small, typically less than 2%.

## 5 EXPERIMENTAL RESULTS

### 5.1 Compare to Other Sparse Storage Formats

Table 1 reports the memory footprint when using different sparse matrix storage formats for BFS. In practice, COL is identical for all CSRs and shared among all running processes, while RST should be duplicated for every running process. Especially for large-scale graph tasks running on an HPC system with thousands of running processes available, amplifying the volume of RST thousands of times. Like CSR and its variants, we compare GraphCSR to state-of-the-art CSR-like formats and consistently outperform all baselines. Specifically, GraphCSR saves more than 90% [4] memory space over most of the CSRs and up to 99.8% of space against the CSCSR format. Furthermore, the average deviation rate in Table 1 is less than 0.4%,

---

[3]The descending order of the RST array prioritizes access to high-degree vertices [23].
[4]Since COL is identical for all CSRs, it is not considered for space saving.

**Table 1: Measured memory footprint ( and difference w.r.t theoretical analysis) per-node for *kron*-31 on 512-node execution.**

| Data Structure | CSR | DCSR | CSCSR | BCSR | GRAPHCSR |
|---|---|---|---|---|---|
| *Auxiliary Arrays* | N/A | 11.74 MB (1.81%) | 764.33 MB (1.81%) | 0.75 MB (0.01%) | 160 B (0.00%) |
| *RST* | 16 MB (0.01%) | 6.41 MB (0.03%) | N/A | 6.41 MB (0.03%) | 1.26 MB (0.06%) |
| *COL* | 48 MB | 48 MB | N/A | 48 MB | 48 MB |
| *TOTAL* | 64 MB (0.01%) | 66.16 MB (1.82%) | 764.33 MB (1.81%) | 55.16 MB (0.03%) | 49.26 MB (0.06%) |

**Table 2: Supercomputer testbed settings**

| Name | #Nodes | CPU | Memory (GB) |
|---|---|---|---|
| *WuzhengLight* | 512 | HG2 64-core CPU | 256 |
| *Tianhe-Exa* | 512 | Phytium 16-core CPU | 16 |

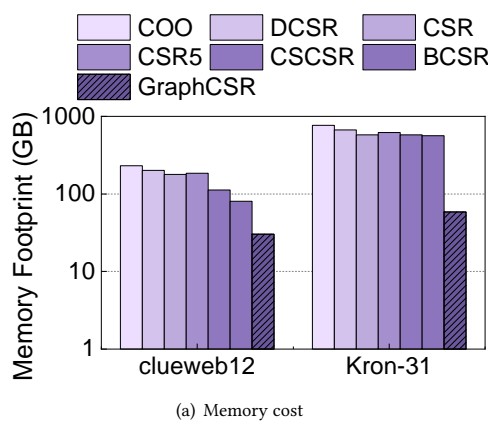

(a) Memory cost

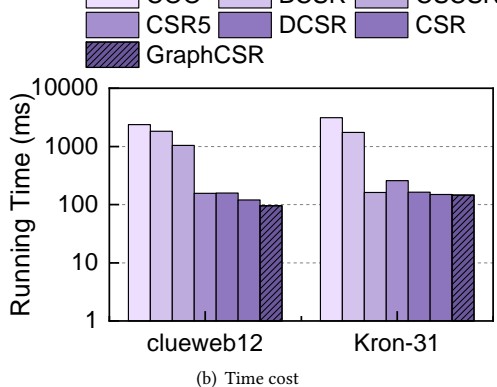

(b) Time cost

**Figure 5: BFS-based GRAPHCSR's (a) Memory and (b) Time consumption against other sparse matrix storage formats.**

even the largest deviation rates remain below 2%, which proves GRAPHCSR's reliability and consistency.

We also evaluate GRAPHCSR on a real-world social graph clueweb12 [1] and report both the memory usage and runtime of BFS shown in Figure 5. Figure 5(a) shows that GRAPHCSR has the smallest memory cost over other storage formats (saving average memory) and in Figure 5(b) GRAPHCSR also shows the fastest running time against others (yielding average runtime speedup). Noted that the time

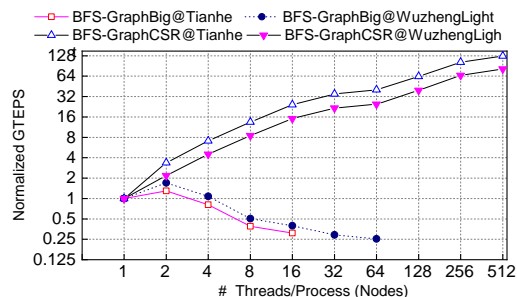

**Figure 6: GRAPHCSR's scalability on Graph500-BFS benchmark.**

**Table 3: GRAPHCSR-based Graph500 BFS vs.Fugaku (the current Graph500 top-ranked supercomputer)**

| System | #Nodes | RAM (GB) | Storage format | GTEPS |
|---|---|---|---|---|
| Tianhe-Exa | 79,024 | 1,264,384 | GRAPHCSR | 162,494 |
| Fugaku | 152,064 | 4,866,048 | BCSR [47] | 137,096 |

cost includes graph preprocessing and all storage formats have implemented the same sorting algorithm before the evaluations.

## 5.2 Scalability

In this section, we evaluate the scalability of GRAPHCSR by applying it to BFS across varying numbers of nodes on WuzhengLight [4] and Tianhe-Exa [55]. Figure 6 reports how the normalized GTEPS changes as we increase the number of computing nodes, using one single node as the normalization baseline. We observe a consistent increase in GTEPS for GRAPHCSR as we increase the number of computing nodes, suggesting that BFS based on GRAPHCSR exhibits good scalability.

## 5.3 GRAPHCSR for Graph500 Ranking

GRAPHCSR has been successfully deployed on the Tianhe-Exa supercomputer using up to 79,024 nodes, where each node is equipped with 16GB of RAM. When running distributed Graph500 BFS using GRAPHCSR on this setup, we achieved a GTEPS of 162,494. This represents a 57.8% improvement over the current top-ranked supercomputer, Fugaku, which delivers a GSETPS of 137,096 using over 148.5K nodes and 3.85× more RAM. As shown in Table 3, the comparison between Tianhe-Exa and Fugaku for Graph 500 BFS demonstrates that our system achieved a higher throughput using fewer computing nodes. These results were obtained on the same graph size generated using the Graph500 graph generation tool, with an edge_factor of 41, resulting in a graph with 2.2 trillion vertices and 35.2 trillion edges. These results confirm the efficiency and

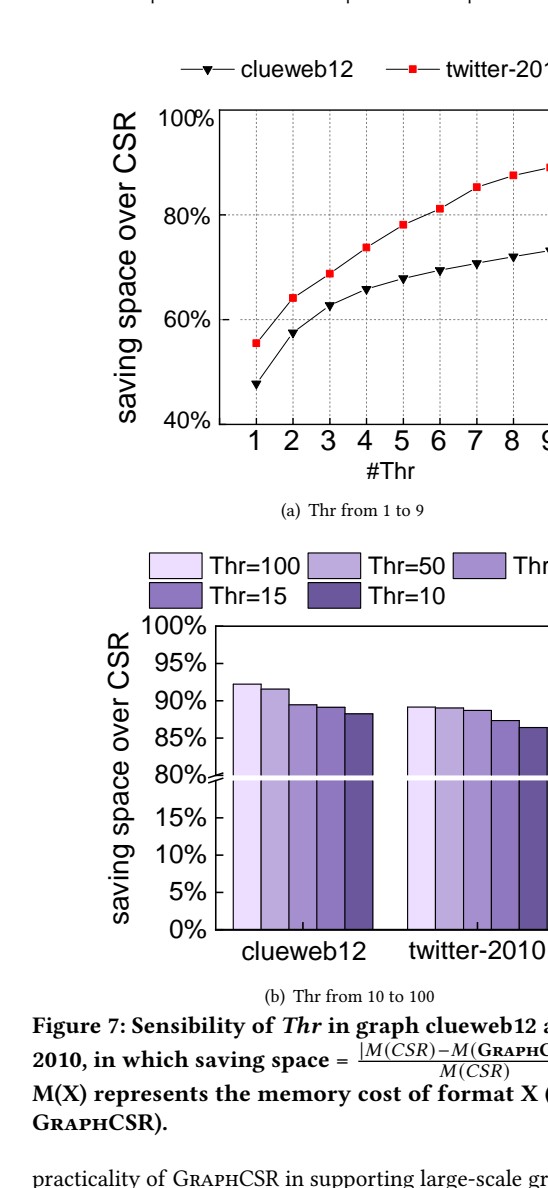

(a) Thr from 1 to 9

(b) Thr from 10 to 100

**Figure 7: Sensibility of *Thr* in graph clueweb12 and Twitter-2010, in which saving space = $\frac{|M(CSR) - M(\text{GRAPHCSR})|}{M(CSR)} \times 100\%$, M(X) represents the memory cost of format X (i.e., CSR or GRAPHCSR).**

practicality of GRAPHCSR in supporting large-scale graph traversal algorithms on modern supercomputers.

## 5.4 Preprocessing overhead

GRAPHCSR requires a sorted graph as input, and further reindex each vertex by the adjunct RST and COL array. This preprocessing is a *one-off* cost. Extensive results show that the processing overhead of GRAPHCSR remains relatively manageable even as the graph size increases (2.07s while scaling to 512 nodes.). We have noticed many great works on vertex sorting in graph-parallel processing systems have been provided, such as [14, 18, 22, 24, 36, 50, 56]. GRAPHCSR is developed for facilitating graph applications, which can seamlessly integrate with existing graph preprocessing approaches while offering additional benefits.

## 5.5 Tuning Edge-degree Parameter

In this subsection, we will take the synthetic graphs generated by the Graph500 data generation tool (Kronecker [28]) to demonstrate how the hyperparameter *Thr* will affect the performance of

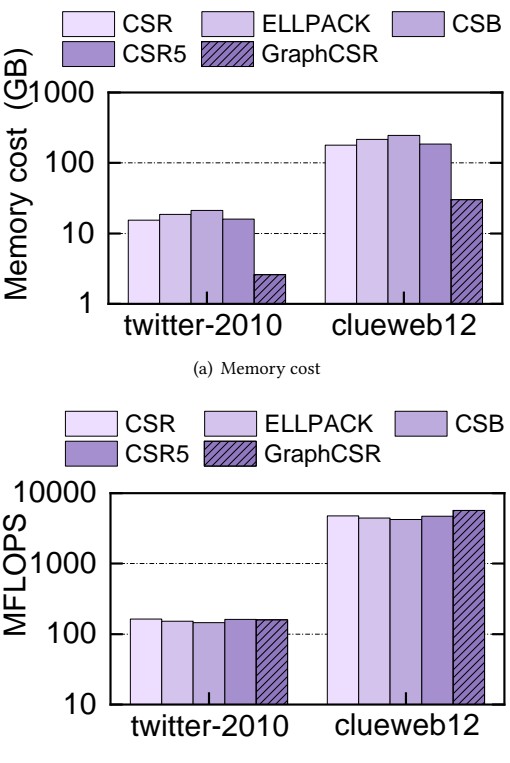

(a) Memory cost

(b) SpMV Throughput

**Figure 8: The SpMV performance of GRAPHCSR with (a) Memory and (b) MFLOPS (Mega FLoating point Operations Per Second) against the state-of-the-art CSR-like formats.**

GRAPHCSR. In addition, we will explore methods for fine-tuning Thr to optimize graph processing speed.

The selection policy for *Thr* should be highly based on the distribution of vertices in the graph. As shown in Figure 3, low-degree vertices constitute a significant portion of real-world graphs, whereas DCSR-mentioned hypersparse graphs are uncommon. Thus, the ideal *Thr* should cover most of the low-degree vertices. For example, since vertices with *degree* $\in [1, 9]$ hold more than 78% for the provided real-world graphs (see in Figure 3), the optimal *Thr* should theoretically be larger than 9. But still, the performance may vary depending on changes in the proportion of low-degree vertices. We highly recommend users to define their own *Thr* range and evaluate the sensibility of *Thr* across different scales of graphs, see Figure 7 according to the graph degrees' distribution.

We evaluate the GRAPHCSR's performance by carefully tuning the *Thr* $\in \{10, 15, 20, 25\}$. We also list the results of *Thr* $\leq 9$ (see Figure 7(a)) to prove that based on the degree distribution in Figure 3, every increase in *Thr* brings obvious benefits, and the overall yield is linear. On the other hand, Figure 7(b) shows that when *Thr* $> 9$, further changes in *Thr* have little effect on its performance with the same graph. The largest performance gap would be around 5% when we conduct different *Thr* on clueweb12 and twitter-2010. So far, we can draw a conclusion that: (i) Majorities of the graphs have a large scale of N-degree vertices, such that $N \leq 10$. In this case, *Thr* = 9 gains significant benefits. First refering to the graph

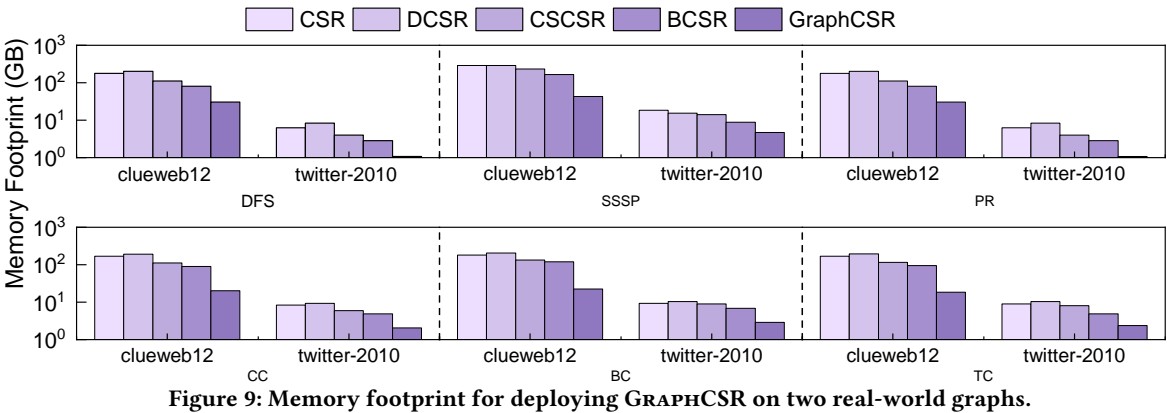

Figure 9: Memory footprint for deploying GraphCSR on two real-world graphs.

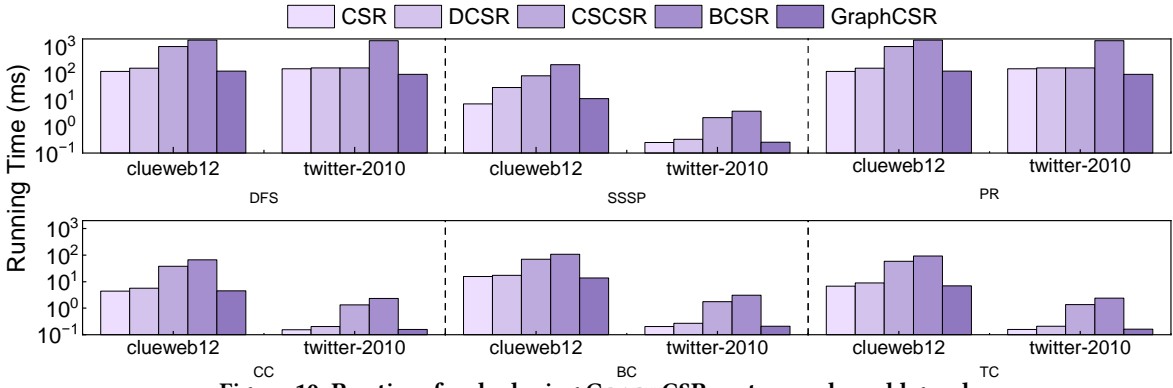

Figure 10: Runtime for deploying GraphCSR on two real-world graphs.

degree distribution before finetuning *Thr* is highly recommended. (ii) Although a larger *Thr* may give a better GraphCSR performance, GraphCSR is overall *Thr*-oblivious when *Thr* > 10.

## 5.6 SpMV Performance

So far, our evaluation is based on graph traversing algorithms like BFS. However, it is noteworthy that GraphCSR can also effectively support SpMV-based graph processing. In this experiment, we evaluate GraphCSR in an isolated SpMV test performed on two real-world datasets: twitter-2010 and clueweb12. We reuse the two input graphs' topology with the randomly generated index vectors as the testing scenario for a fair comparison. Figure 8 shows the SpMV performance of the CSR-based variants, including ELLPACK [29], CSB [7], and CSR5 [37].

Figure 8(a) shows that GraphCSR yields lower memory cost against all prior CSR-like formats by up to orders of magnitude. On average, those CSR-based formats require extra memory to support vectorization and tiling, which perform well in small graph computations but are fatigued when facing large graphs. By considering memory efficiency, our approach outperforms the state-of-the-art storage formats like ELLPACK, CSB and CSR5 as shown in Figure 8 (b).

## 5.7 GraphCSR for Real-World Graphs

We conducted experiments with GraphCSR on large real-world graphs, including clueweb12 and twitter-2010, using various graph algorithms, such as DFS, SSSP, PR, CC, BC, and TC, based on different CSR-like formats, as shown in Figure 9 and Figure 10. The results show that GraphCSR outperforms prior CSR-like formats on all datasets and evaluation metrics. GraphCSR surpasses all the CSR formats by saving up to 89.2% and 71.9% (average 77.3% and 65.5%) of space compared to naive CSR and BCSR, respectively. And refer to Figure 10, GraphCSR outperforms all the CSR formats and offers at most 19.3 speedups (average 14.4×) while running each popular graph algorithm. It's worth noticing that since real-world graphs are typically not hypersparse, DCSR requires more memory than BCSR or even vanilla CSR. Correspondingly, our approach demonstrates high stability when dealing with both hypersparse and non-hypersparse graphs since we do not solely rely on the number of 0-degree vertices as DCSR does.

## 6 CONCLUSION

We have presented GraphCSR, a CSR-like sparse storage format designed for large-scale graph applications in web-scale environments, where memory efficiency is a significant concern. GraphCSR leverages the observation that most of the vertices of real-world graphs have low edge degrees, and many vertices with the same edge degree can be grouped to reduce memory consumption in storing the graph adjacency matrix. Through both theoretical analysis and empirical evaluation on two high-performance clusters, we demonstrated that GraphCSR consistently outperforms existing sparse matrix storage formats across a variety of graph operations. By delivering higher throughput and reduced memory consumption, GraphCSR enhances the efficiency of large-scale graph processing in distributed high performance computing environments, making it particularly suitable for modern web, mobile, and WoT systems central to the Web's infrastructure.

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
