# OpenReview forum: "GraphCSR: A Space and Time-Efficient Sparse Matrix Representation for Web-scale Graph Processing"
_ACM.org/TheWebConf/2025/Conference — WWW 2025 Oral_

### Official Review · Reviewer_vdVf · 2024-11-29

**Novelty:** 4
**Technical Quality:** 4

**Review:**

The GraphCSR method proposed in this paper is based on two key observations from CSR: "arrays are sorted in non-decreasing order" and "vertices with the same edge degree have their adjacent lists stored contiguously in the COL array." It aims to improve storage efficiency in web-scale graph processing, demonstrating both innovation and practical application potential. However, there are several shortcomings in the paper: First, the method description is somewhat brief, lacking sufficient explanation of the design choices and a clear discussion of its innovation. Second, the experimental section lacks necessary details, including dataset selection, experimental setup, and comparisons with existing methods. Finally, the discussion of experimental results is somewhat superficial and does not fully highlight the advantages of GraphCSR in different scenarios. It is recommended that the authors further clarify the innovation of the method and provide more detailed experimental data and result analysis to strengthen the paper's contributions.
Pros:
1.The proposed GraphCSR method has high practical value in improving storage efficiency for web-scale graph processing, especially in the context of increasing graph sparsity and the need for large-scale parallel computing.
2.While there is room for improvement in the implementation, the idea of sorting vertices by degree and applying compression offers an innovative approach that theoretically enhances storage efficiency.
3.Extensive experiments are conducted to verify the effectiveness of each proposed component in the framework.

Cons:
1.The overall structure and logic of the article are somewhat confusing, making it hard to follow. There is a lot of redundant and repetitive expression, with many sections emphasizing the significance of the proposed approach but failing to clearly explain how it works. The paper reads more like a survey than a research paper, lacking enough details and clear step-by-step descriptions.
2.In Section 3, while the consideration of vertices with the same degree is introduced, compared to traditional methods, only two additional arrays are introduced for representation, which lacks significant innovation. Additionally, the paper does not sufficiently explain the necessity of this design and how it improves upon existing methods.
3.In Section 3.2, the paper uses a simple example to explain what low_deg_RST and low_deg_COL store, but there is a lack of detailed description of the overall method. While the example helps with understanding, it still does not provide enough technical detail to fully grasp the method.
4.The experimental section does not clearly define the baselines, and the overall experiment is quite disorganized. The results are not discussed in sufficient detail. Although the paper aims to address the limitations of sparse matrix storage efficiency, the experimental discussion is quite brief, with more focus on parameter settings rather than results analysis.

**Questions:**

1.The introduction section focuses on background information but lacks a clear motivation. Is the goal to improve storage efficiency? Why can't traditional methods achieve this? What are the limitations or challenges the proposed method addresses?
2.Section 2 provides a lot of redundant background information. The key points, such as how CSR represents graph structures and what Web-scale graphs are like, could be summarized more concisely without excessive elaboration. This part overlaps with the introduction and should be more focused.
3.The second paragraph mentions two observations based on RST—"arrays are sorted in non-decreasing order" and "vertices with the same edge degree have their adjacent lists stored contiguously in the COL array." What can be done with these observations? These observations should provide the motivation for the method, but this isn't clearly explained. This section should be expanded to clarify the motivation behind these observations.
4.The experiment in Section 5.1 does not specify which datasets were used.
5.Each experiment should include basic descriptions to make it clearer for the reader, such as what is being verified, which dataset was used, what baselines were compared, and what conclusions were drawn.

**Reviewer Confidence:**

2: The reviewer is willing to defend the evaluation, but it is likely that the reviewer did not understand parts of the paper

**Scope:**

3: The work is somewhat relevant to the Web and to the track, and is of narrow interest to a sub-community

---

### Official Review · Reviewer_dXM8 · 2024-12-01

**Novelty:** 6
**Technical Quality:** 6

**Review:**

This paper presents GraphCSR, an innovative sparse matrix storage format tailored for large-scale graph processing. The authors provide comprehensive experimental results comparing GraphCSR to existing CSR-like formats across various graph traversal algorithms and real-world datasets. The results indicate significant improvements in both memory efficiency and execution speed on HPC systems, particularly for BFS and SpMV operations.

### Pros:
- GraphCSR achieves substantial memory savings while maintaining high processing speed, making it ideal for large-scale graph applications.
- The use of industry-standard benchmarks (Graph500) and real-world graphs strengthens the credibility of the results.
- Strong performance across varying node configurations demonstrates the approach’s scalability for large supercomputing environments.
- The integration of real-world datasets like clueweb12 and twitter-2010 shows that GraphCSR is not just limited to synthetic workloads.

### Cons:
- While the preprocessing time is low, sorting and reindexing the graph before processing can be an additional burden in certain workflows.
- The performance is influenced by the choice of Thr, and while the paper suggests optimal values, a more thorough exploration of this parameter across various graph structures would provide clearer guidelines for users.
- The paper primarily focuses on BFS and SpMV, leaving other graph algorithms (like SSSP, PR, etc.) less explored in terms of performance comparison.
- The effectiveness of Thr seems to depend heavily on the degree distribution of the graph, which might not be as predictable in dynamic, real-world graphs.

### Originality and Significance:
The work is **highly original** in introducing a new sparse matrix format designed for graph processing that balances memory efficiency and execution speed. Given the increasing importance of large-scale graph analysis in AI and machine learning, the significance of this research lies in its potential to optimize the memory usage and performance of graph processing tasks, especially on supercomputing platforms. The paper’s contribution to the field of **high-performance graph analytics** is significant, especially in the context of increasingly large graphs encountered in social networks, web data, and scientific computing.

### Writing Style and Presentation:
- The overall structure and clarity of the paper are strong. However, certain sections, such as the methodology and algorithm comparison, could benefit from additional clarifications. For example, while the results are clearly presented, the authors could expand on why GraphCSR performs better in specific scenarios and how this relates to the characteristics of the datasets used.

- The paper could go deeper into explaining the trade-offs between GraphCSR and other sparse formats in terms of the preprocessing cost and memory layout. While the authors mention Thr's impact on performance, a more detailed discussion on how Thr interacts with graph structure would be valuable.

- While the figures and tables provide useful insights, some of them (especially those comparing memory and time costs) could be more effectively labeled with clearer captions to improve readability. For example, the use of subplots (e.g., Figure 7) can sometimes be overwhelming. Simplifying the figures and providing more context in the captions would help the reader digest the results more easily.

- The mathematical notation is generally clear, but a few key concepts (such as the definition of Thr) would benefit from a more formal description early in the paper. Equations are presented clearly, but more elaboration on their significance in the context of GraphCSR's design could enhance the paper’s accessibility.

**Questions:**

1. Could the authors provide further analysis on how Thr affects performance across a broader range of graph algorithms? Specifically, does adjusting Thr lead to consistent benefits in algorithms other than BFS and SpMV?
2. While GraphCSR outperforms other CSR-like formats, how does it compare with DCSR in handling highly sparse graphs? Are there specific types of sparse graphs where GraphCSR might struggle or underperform?
3. Can the authors quantify the impact of the preprocessing step (sorting and reindexing) on overall system performance for larger graphs? Does this overhead remain manageable as graph sizes scale up further?
4. The scalability of GraphCSR is demonstrated for BFS. How does GraphCSR’s performance scale with other complex graph algorithms such as SSSP or PR when applied on real-world datasets like clueweb12 or twitter-2010?
5. Given that real-world graphs are often dynamic and evolve over time, how well does GraphCSR handle dynamic updates, such as vertex or edge insertions, compared to static graphs?
6. The paper mentions that GraphCSR is less reliant on the number of 0-degree vertices compared to DCSR. Can the authors elaborate on how GraphCSR handles hypersparse graphs with many isolated nodes, and whether performance is affected in such scenarios?

**Reviewer Confidence:**

4: The reviewer is certain that the evaluation is correct and very familiar with the relevant literature

**Scope:**

3: The work is somewhat relevant to the Web and to the track, and is of narrow interest to a sub-community

---

### Official Review · Reviewer_WcBw · 2024-12-01

**Novelty:** 7
**Technical Quality:** 7

**Review:**

Summary:
In this paper, the authors introduce GraphCSR, a novel compact data structure for large graphs that extends the traditional Compressed Sparse Row (CSR) format. The key insight behind GraphCSR is that many graph vertices exhibit similar low-edge degrees, which allows for effective grouping. This grouping significantly reduces both storage requirements and runtime memory consumption, making it an efficient solution for large-scale graph representation.

Strong Points:

S1: The paper addresses a fundamental problem in graph representation for large graphs, offering a valuable contribution to the field.
S2: The system is evaluated on graphs and distributed computing setups that are much larger in scale compared to most existing studies. The evaluation is rigorous, employing the widely recognized Graph 500 benchmark to validate the results.

**Questions:**

W1: The proposed GraphCSR data structure could benefit from a more formal description, along with a detailed discussion of its complexity.

**Reviewer Confidence:**

4: The reviewer is certain that the evaluation is correct and very familiar with the relevant literature

**Scope:**

4: The work is relevant to the Web and to the track, and is of broad interest to the community

---

### Official Review · Reviewer_3qiM · 2024-12-02

**Novelty:** 4
**Technical Quality:** 5

**Review:**

GraphCSR has clear applicability in various large-scale graph processing scenarios, such as social networks, recommendation systems, and the Internet of Things, particularly demonstrating practical value in distributed systems and edge computing environments. The article comprehensively evaluates the performance advantages of GraphCSR through experiments on various algorithms (e.g., BFS, SpMV) and commonly used large-scale graph datasets, clearly showing significant improvements in memory usage and execution speed compared to multiple mainstream storage formats.

Comments：
1.Although the core idea of GraphCSR is based on degree-based vertex grouping, the description of its internal implementation details (such as grouping methods and boundary condition handling) is somewhat vague, particularly regarding the transition between handling low-degree and high-degree vertices, lacking more precise algorithm details and pseudocode.
2. While the article compares eight sparse matrix storage formats, it does not provide detailed criteria for selecting these baseline methods and does not address some of the latest dynamic graph storage methods or other optimized storage formats for graph processing (such as those based on graph databases).
3. The article mentions that GraphCSR introduces a key parameter, Thr (used to distinguish between low-degree and high-degree vertices), but does not delve into the rationale for choosing this parameter, its specific impact on performance, or its applicability in different graph structures (for example, what threshold values are suitable for dense and sparse graphs, respectively).
4. Although experiments were conducted in a supercomputing environment, there is limited testing on edge computing and resource-constrained devices, failing to adequately reflect the advantages of GraphCSR in these scenarios.
5. The article, while demonstrating the performance of GraphCSR on supercomputers, offers little discussion on its scalability in other distributed environments (such as cloud computing platforms), particularly lacking an analysis of communication overhead.
6. Figure 3 presents the degree distribution of different datasets but does not discuss in detail how these distribution characteristics specifically affect the performance of GraphCSR, such as changes in storage efficiency under different distributions.

**Questions:**

1. How is the parameter Thr introduced in GraphCSR determined? Does the choice of this parameter affect performance in different graph structures?
2. How does GraphCSR perform in a supercomputing environment? What is its scalability like in other distributed computing environments?
3. How does GraphCSR handle low-degree and high-degree vertices? What challenges might arise in this process?
4. What is the impact of the degree distribution characteristics shown in Figure 3 on the performance of GraphCSR? How does storage efficiency vary under different distributions?

**Reviewer Confidence:**

2: The reviewer is willing to defend the evaluation, but it is likely that the reviewer did not understand parts of the paper

**Scope:**

3: The work is somewhat relevant to the Web and to the track, and is of narrow interest to a sub-community

---

### Official Review · Reviewer_kioU · 2024-12-03

**Novelty:** 6
**Technical Quality:** 6

**Review:**

The authors address the problem of efficiently storing large-scale graphs. As such, they propose the novel GraphCSR sparse storage format that builds on the CSR format and grouping and indexing vertices based on their degree. The performance benefits of the approach are experimentally assessed against those of state-of-the-art baselines, on both real-world and synthetically generated data, obtained by levering the Graph500 benchmark.


Strong points:
1. The problem of storing large-scale graphs efficiently with a reduced memory footprint is important and practical for real world web applications.
2. The authors propose a novel format built on top of the popular CSR format and experimentally demonstrate its superiority compared to state-of-the-art baselines.
3. The approach is compared against multiple baselines on datasets of large-scale datasets of (hundreds of) millions of nodes and billion edges.

Weak points:
1. The figures in the experimental study section can be refined. The color set of Figure 5, Figure 7, and Figure 8 are not friendly to the readers, distinguishing different methods through the darkness of purple is hard and may lead to misreading. Suggest to use obviously different colors for different methods. In addition, Figures 5~8 can be scaled smaller.
2. Some format issues should be fixed. For example, line 632 on page 6 goes across the middle margin.
3. The experimental study section shows that the proposed GraphCSR can really reduce the memory usage of storing large-scale graphs, however it is better to give some space cost analyses in section 3 to make this study more solid.

**Questions:**

The scope of application of GraphCSR is a sorted graph (the row offset of CSR is arranged in a Non-descending order). Can SuperCSR be promoted and applied in cases where it is not applicable to sorted graphs? For example, when converting an unsorted graph to a sorted graph, can algorithmic optimization be carried out to expand the applicability of GraphCSR?

**Reviewer Confidence:**

4: The reviewer is certain that the evaluation is correct and very familiar with the relevant literature

**Scope:**

4: The work is relevant to the Web and to the track, and is of broad interest to the community